# Development and Implementation of an Advanced Program for Robotic Treatment of Prostate Cancer—Is Surgical Quality Transferable?

**DOI:** 10.3390/cancers14215261

**Published:** 2022-10-26

**Authors:** August Sigle, Cordula A. Jilg, Moritz Weishaar, Boris Schlenker, Christian Stief, Christian Gratzke, Markus Grabbert

**Affiliations:** 1Department of Urology, Faculty of Medicine, Medical Centre, University of Freiburg, Hugstetter Str. 55, 79106 Freiburg, Germany; 2Berta-Ottenstein-Programme, Faculty of Medicine, University of Freiburg, 79085 Freiburg, Germany; 3Department of Urology, Ludwig-Maximilians-University of Munich, 80539 Munich, Germany

**Keywords:** prostatic neoplasm, prostatectomy, robotics, treatment outcome, postoperative complications, implementation science

## Abstract

**Simple Summary:**

Robot-assisted radical prostatectomy (RARP) is a surgical treatment option for prostate cancer. The quality of this procedure depends on the surgeon’s operative volume and expertise. When implementing RARP at a new center, it is standard practice to hire a surgeon with appropriate specialty training and expertise. However, since factors other than the surgeon may play a role, the aim of this study was to evaluate the transferability of quality. We compared two different cohorts operated on by the same surgeon who offered extensive training and experience. When analyzing relevant outcome parameters such as duration of surgery, blood loss and the oncologically relevant proportion of positive surgical margins, we found that the results of the second cohort were comparable to those of the first. Thus, we conclude that the quality of RARP is transferable if a surgeon with extensive specialist training and expertise is hired.

**Abstract:**

Introduction: Robot-assisted radical prostatectomy (RARP) is a surgical treatment option for prostate cancer (PC). Quality in RARP depends on the surgeon´s operative volume and expertise. When implementing RARP, it is standard practice to hire a pre-trained surgeon. The aim of our study was to investigate the transferability of quality in RARP. Patients and Methods: We analyzed two consecutive retrospective cohorts of 100 and 108 men, respectively, who underwent RARP at two different centers and on whom surgery was performed by the same surgeon. Results: There were more men with high-grade PC in Cohort 1: 25/100 (25.0%) vs. 9/108 (8.3%), *p* < 0.01, and infiltration of the seminal vesicles was more frequent (23/100 (23.0%) vs. 10/108 (9.2%), *p* < 0.01). In Cohort 2, the duration of surgery was shorter and blood loss was lower: 149 (134–174) vs. 172 min (150–196), *p* < 0.01 and 300 (200–400) vs. 131 (99–188) mL, *p* < 0.01. No difference was found in the proportion of positive surgical margins in the T2 cohort (8.8% vs. 8.2%, *p* = 1.00). Conclusion: The procedural and oncological outcome parameters of Cohort 2 do not appear to be inferior to the results obtained for the first cohort. The quality of RARP is transferable if a pre-trained surgeon is hired.

## 1. Introduction

Robotic-assisted surgery is a rapidly evolving field. An increasing number of hospitals is implementing this forward-looking surgical option in their clinical routine including a broad variety of disciplines such as gynecology, urology, thoracic, general and head and neck surgery [1,2]. In the field of urology, robotic surgery has become the standard of care for certain indications, as around 80% of radical prostatectomies in the US are performed robotically assisted [3]. Robot-assisted radical prostatectomy (RARP) is a standardized surgical treatment option in patients with prostate cancer (PC). Because the lower pelvis is easily accessible with the DaVinci robotic system (Intuitive Surgical Inc., Sunnyvale, CA, USA), this procedure has been successfully adopted in the surgical treatment of PC, with a growing number of systems and procedures worldwide [4].

The surgical quality, including functional outcome parameters, of RARP has been shown to be dependent on the surgeon´s operative volume and expertise, and seem to be slightly superior to the open approach because it has better teaching capabilities [5]. Several studies have demonstrated learning curves for various outcome parameters, with a plateau reached after 20–300 cases for operative time, blood loss and positive surgical margins (PSM) rates [1]. To bypass these learning curves, a common approach to introducing RARP to a new center is to hire a well-trained surgeon, who is not only implementing a standardized set up but is also bringing the appropriate surgical experience needed. However, factors other than surgeon experience may affect outcome parameters when implementing RARP at a new center. 

Currently, there is a clear lack of evidence regarding the transferability of quality in RARP. The aim of our study was to compare procedural and oncological outcomes of two consecutive cohorts of a single surgeon (C.G.) at two German centers. Since December 2018, RARP has been performed regularly at University Hospital Freiburg (UHF). The surgeon implementing this procedure was trained at University Hospital Munich (UHM) and OLV Aalst and performed more than 500 RARPs at his former institution.

## 2. Patients and Methods

### 2.1. Study Population

We analyzed two consecutive retrospective cohorts of men who underwent RARP at two different academic centers (UHM (Cohort 1) and UHF (Cohort 2)) between May 2018 and September 2018 and December 2018 and December 2019, respectively. Both cohorts were operated on by the same surgeon (C.G). Cohort 1 was the last consecutive cohort of men that was operated on at UHM. Vice versa, Cohort 2 comprised the first consecutive cohort from UHF. Retrospective data extraction was performed using a predefined data dictionary. For both cohorts, the indication for RARP was based on biopsy-confirmed diagnosis of PC. Men who had had any previous therapy for PC were excluded from our analysis. The initial data set included a total of 102 (Cohort 1) and 111 men (Cohort 2), respectively. Two men were excluded, because they had not undergone RARP (error in the initial data extraction); one had had prior radiotherapy of the prostate; and two had incomplete clinical data. The final data set included 100 men in Cohort 1 and 108 men in Cohort 2. Data collection was approved by the local Ethics Committee (Lead Investigating Center, UHF: ETK 21-1439, 22 July 2021). The study was performed in accordance with the Declaration of Helsinki.

### 2.2. Robotic-Assisted Radical Laparoscopic Prostatectomy

RARP was performed on a DaVinci Xi and a DaVinci X system in the four-arm configuration and via a transperitoneal approach. The patient was placed in supine position and the arms were attached laterally to both sides. A transurethral catheter was inserted. For the establishment of the pneumoperitoneum a longitudinal incision of approximately 3 cm above the umbilicus was used. The fascia incision and access to the abdomen were performed under visual control and holding sutures were laid at the lower and upper end of the external rectus fascia. After the first trocar was inserted, the peritoneal space was insufflated with carbon dioxide. The trocars were positioned following an arcuate line: on the right side one DaVinci trocar (8 mm, medioclavicular line) and one additional trocar for the assistant (12 mm, anterior axillar line). Two DaVinci trocars were inserted on the left side (8 mm, medioclavicular and anterior axillar line). Thereafter, the patient was positioned in deep Trendelenburg position and the DaVinci system was connected. A fenestrated bipolar forceps, a ProGrasp forceps, a needle driver and a pair of monopolar scissors were used. Surgery was performed in a standardized descending technique. In the first step, the bladder was detached from the abdominal wall following the avascular layer. Hereafter, the anterior surface of the prostate was exposed. The fatty tissue was completely removed and sent to pathology. The puboprostatic ligaments on both sides were exposed and the endopelvic fascia was incised from dorsal to ventral followed by the dissection of the bladder neck. After opening the bladder neck, the catheter was used to keep the prostate in position. Thereafter, the posterior portion of the bladder neck was transected and the base of the prostate was detached from the posterior wall of the bladder. In the next step, the deferent ducts on both sides were identified, dislocated, clipped and transected. The seminal vesicles were visualized and mobilized. Seminal vesicle arteries were clipped. Denonvilliers’ fascia was incised carefully, the rectum was pushed away meticulously and the lateral pillars of the prostate were exposed. Lateral to the prostate, the neurovascular bundles were detached from the gland and the remaining prostatic pillars were then gradually detached with clips. The Santorini plexus was incised without any prior sutures for ligation and the urethra was severed without electricity. A sufficient urethral stump was left in place. At this time, the specimen was evacuated via an endo pouch and sent for rapid pathological analysis of the gland. Lymph node dissection (LND) was performed according to pre-described templates, whereby the extent of LND was based on preoperative risk classification [2]. In Cohort 2, all patients underwent LND. In Cohort 1, LND was omitted in 5 low-risk patients. Anastomosis was performed by posterior musculofascial reconstruction with approximation of Denonvilliers’ fascia to the posterior aspect of the sphincteric complex, followed by continuous suturing of the anastomosis using a previously described technique [3]. After this step, a 20 Charriere catheter was inserted and the anastomosis was tested with retrograde filling of the bladder. Then, subtle control of the surgical field was performed and the trocars were removed under vision. At the same time, a drain was inserted via the right lateral DaVinci trocar. In a final step, the abdominal wall was reconstructed anatomically and the skin was closed with intracutaneous sutures. Before the removal of the catheter, anastomosis was tested by cystography.

### 2.3. Implementation of the Surgical Procedure and Standardized Set-Up

The first surgeries were prepared in an interdisciplinary meeting between surgeons, scrub-nurses and anesthesiologists. A physician who had also been trained externally acted as bedside assistant for the first 50 cases. The scrub-nurses were introduced to the new technique in several structured meetings on site. In addition, they were trained at a specialized training center and visited a department with broad experience in robot-assisted surgery prior to the first case. The first patient was selected carefully and after an interdisciplinary discussion to reduce the risk of intraoperative complications. The system was tested several times in the very same OR to identify possible pitfalls. The day before the first surgery the team met for a dry run. Even experienced surgeons who have previously worked in specialized hospitals may underestimate the importance of this targeted preparation for their first cases. After initial implementation of the system and set-up, additional nurses and bedside assistants, as well as console surgeons, were trained in a step-by-step approach.

### 2.4. Pathological Work-Up of Specimen

A structured frozen-section examination was performed during surgery with analysis of both neurovascular bundles and the apex. The entire specimen was then examined by a group of experienced uropathologists according to a standardized protocol in 0.5 cm slices to minimize false-negative results for PSM.

### 2.5. Data Collection and Statistical Analysis

Demographic and clinical data were extracted by reviewing the patients’ electronic medical records. Baseline characteristics included age, body mass index, prostate-specific antigen (PSA), prostate volume derived from the radical prostatectomy specimen (weighting of the surgical specimen) and the histopathological results of the prostate biopsy including Gleason score, total number of biopsy cores taken, and the number and proportion of positive cores.

Surgical outcome parameters included the Gleason score of the radical prostatectomy specimen, pathological T stage, pathological N stage, surgical margin status, duration of surgery (total operative time) and estimated blood loss. For Cohort 1, blood loss was estimated by the operating surgeon. For Cohort 2, blood loss was calculated by multiplying the perioperative difference in hemoglobin by the patient’s estimated blood volume (70 mL/kg bodyweight).

Concerning adverse events and hospital readmission rates, we analyzed intraoperative complications and readmission rates within 30 and 90 days, respectively. Readmission included all patients with unscheduled hospitalization. Complications were available for the period of hospitalization and reported according to the Clavien–Dindo classification (CDC) [4]. Acute pulmonary insufficiency (measured oxygen saturation <90% and application of oxygen), acute kidney impairment (rising creatinine level > 0.3 mg/dL within 48 h) and disorders of the liver (2-fold elevation in liver enzyme levels) were classified as CDC grade 1. Infectious conditions treated with an antibiotic were classified as CDC grade 2, and interventions without the need for general anesthesia such as drainage of a lymphocele or urinoma were classified as grade 3a. Continuous variables were described as median with interquartile range (IQR). Categorical variables were described with integers and percentages. For analysis of the differences between the two cohorts, we applied the Mann–Whitney *U* test for comparison of continuous variables and the chi-squared test for the categorical variables. A *p*-value < 0.05 was considered statistically significant. SPSS© software (SPSS statistics Version 27, IBM, Armonk, New York, NY, USA) was used for statistical analyses.

## 3. Results

### 3.1. Baseline Characteristics

The baseline characteristics of the two cohorts are presented in Table 1. The median (IQR) PSA and age for Cohort 1 vs. Cohort 2 were 65.0 (60.0–71.0) vs. 65.0 (58.3–70.0) years, and 8.5 (5.7–12.4) vs. 8.7 (5.5–12.0) ng/mL, respectively. The differences did not show statistical significance. The prostate volumes (IQR) in Cohort 1 were significantly larger as compared to Cohort 2: 54.0 (43.0–65.0) vs. 47.0 (37.0–60.0), *p* = 0.01. Moreover, there was a significant difference in the histopathological results of the prostate biopsy (*p* < 0.01), with a greater proportion of high-grade PC in Cohort 1: 27/100 (27.0%) vs. 6/108 (5.6%).

### 3.2. Outcomes of the Robot-Assisted Radical Prostatectomy

The differences in the histopathological results of the prostate biopsy were consistent when analyzing the outcomes of the radical prostatectomy, with significantly more high-grade PC found in Cohort 1: 25/100 (25.0%) vs. 9/108 (8.3%), *p* < 0.01 There was also a difference in the proportions of the pathological T stages (*p* < 0.01), with more frequent infiltration of the seminal vesicles in Cohort 1 (23/100 (23.0%) vs. 10/108 (9.2%)). Significantly shorter duration of surgery and lower estimated blood loss was noted for Cohort 2: 149 (134–174) vs. 172 min (150–196), *p* < 0.01 and 300 (200–400) vs. 131 (99–188) mL, *p* < 0.01. No differences were found in the proportions of PSM with respect to oncological outcome, neither in the T2 cohort (8.8% vs. 8.2%, *p* = 1.00), nor in the total cohort (18.0% vs. 16.7%, *p* = 0.751) (Table 2).

### 3.3. Hospital Readmission Rates and Adverse Events

Hospital readmission rates and adverse events for Cohort 2 are shown in Table 3. Data on these aspects were not available for Cohort 1. No intraoperative complications were reported. The 30- and 90-day hospital readmission rate was 3/108 (2.8%) and 5/108 (4.6%), respectively. Three men were hospitalized due to a lymphocele and treated with drain placement; one man presented with an unclear sepsis that was treated with antibiotics and there was one case of unclear vertigo that resolved spontaneously.

According to the CDC, 13/108 (12.0%), 20/108 (18.5%) and 5/108 (3.7%) adverse events occurred classified as grade 1, grade 2 and grade 3a, respectively. No higher-grade complications were reported. Grade 2 complications were mostly postoperative bacteremia routinely treated with antibiotic therapy. Grade 3a complications included the presence of a lymphocele (3/108, 2.8%), one case of a urinoma (0.9%) requiring drain placement and one case of urinary retention (0.9%) requiring transurethral catheterization. No patient needed a blood transfusion, and no revision surgery was performed.

## 4. Discussion

Robotic-assisted laparoscopic radical prostatectomy has attained widespread diffusion and adoption [5]. Quality in RARP has been shown to be dependent on surgeon expertise, with different learning curves for different outcomes and a plateau for operative time, blood loss and PSM rate after 20–300 cases [1]. To bypass these learning curves, when introducing RARP to a new center, it is standard procedure to hire a surgeon with the appropriate specialist training and expertise. However, when implementing RARP in a new hospital, factors other than surgeon experience alone may influence outcome parameters. Currently, there is limited evidence on the transferability of quality in RARP. The aim of our study was to compare procedural and oncological outcomes of two consecutive cohorts at two German centers that were operated on by the same surgeon.

### 4.1. Outcomes of the Robot-Assisted Laparoscopic Radical Prostatectomy

Firstly, we analyzed procedural outcomes of the RARP. For Cohort 2, blood loss (IQR) was significantly lower compared to Cohort 1: 300 (200–400) vs. 131 (99–188) mL, *p* < 0.01. These differences might originate from different methods of determination: for Cohort 1, blood loss was estimated by the operating surgeon and for Cohort 2 it was calculated based of the perioperative differences in hemoglobin concentrations. Our findings are in accordance with the results of a large systematic review including 110 studies that found a mean blood loss of 166 mL [6]. A significantly shorter duration of surgery was reported for Cohort 2: 149 (134–174) vs. 172 min (150–196), *p* < 0.01. Overall, this procedural duration is in line with previous studies [6]. The differences between the two cohorts might be attributable to the higher proportion of high-grade and locally advanced tumors in Cohort 1 and are, thus, unlikely to be a consequence of a further operator learning curve. At the very least, this result can be interpreted as an indication that other effects besides surgeon experience do not significantly influence the duration of surgery when RARP is introduced to a new center.

Regarding oncological outcome, we found no differences between the two cohorts in the proportion of PSM neither in the T2 cohort (8.8% vs. 8.2%, *p* = 1.00), nor in the total cohort (18.0% vs. 16.7%, *p* = 0.751). These rates are comparable to previously reported results from two different systematic reviews that found PSM rates of 17.6% and 15.0%, respectively [7,8]. Again, this result can be interpreted as an indication that there are no major factors affecting the oncologic outcomes of the procedure other than the surgeon´s track record. For the training of a novice in RARP, mentoring programs seem to represent the most favorable approach, as a recent article showed that trainees seem to reach their turning points in learning curve after far fewer cases than their mentors [9]. Moreover, another recent study found that trainees duplicate their respective mentors’ outcomes during learning curve [10]. This leads to the conclusion that the training in RARP should be centralized in high-volume centers with excellent outcome parameters.

### 4.2. Hospital Readmission Rates and Adverse Events

For the second consecutive cohort (Cohort 2, UHF), we analyzed hospital readmission rates and adverse events. For hospital readmission rates, we found 3/108 (2.8%) and 5/108 (4.6%) readmitted after 30 and 90 days, respectively. These rates are consistent with a recently published review article that included more than 400.000 patients over a 20-year period and found a 30-day postoperative readmission rate of 4% [11]. With regard to adverse events according to the CDC, 13/108 (12.0%), 20/108 (18.5%) and 5/108 (3.7%) were classified as grade 1, grade 2 and grade 3a, respectively. Grade 3a complications included the presence of a lymphocele (3/108, 2.8%), one case of a urinoma (0.9%) requiring drain placement and one case of urinary retention (0.9%) requiring transurethral catheterization. A systematic review of more than 19.000 cases described lower rates of grade 1 and grade 2 complications: 2.1% and 3.9%, respectively [7]. These differences could be due to ambiguous definitions of lower grade complications. In our study, the standards for recording complications were rather strict compared with others. To give two examples: we defined an oxygen saturation level <90% and concomitant application of oxygen as a grade 1 complication, and any bacteremia (even if asymptomatic) treated with an antibiotic as a grade 2 complication. Otherwise, the proportions of grade 3a adverse events are consistent with a high-volume review by Novara et al., who found lymphoceles and urinoma in 3.1% and 1.8% of cases, respectively [6].

### 4.3. Limitations and Strengths of the Study

The present study has several limitations. Firstly, the retrospective design and small sample size limit the generalizability of our results. Secondly, the two cohorts were not perfectly matched, because there were more high-grade and more locally advanced PCs in Cohort 1. This might interfere with the endpoints reported such as duration of surgery, estimated blood loss and proportion of PSM. In addition, the different methods of blood loss estimation in the two cohorts limit the usability of this parameter.

Furthermore, we could not report functional outcomes or information on comorbidities, because no data were available. However, to our knowledge, this is the first study to describe procedural and oncological outcomes from two consecutive cohorts operated on by a single surgeon.

## 5. Conclusions

The outcome parameters, such as the duration of surgery, estimated blood loss and the rate of PSM are consistent with the body of evidence. More importantly, the results from the second cohort (after implementation of RARP at the new center) do not appear to be inferior to the results of the first cohort.

Thus, we conclude that quality in RARP is transferable if a highly trained surgeon with the pertinent expertise in this field is hired. This strategy may avoid deficiencies in patient care because of the learning curves associated with the procedure.

## Figures and Tables

**Table 1 cancers-14-05261-t001:** Baseline characteristics.

Characteristic	Cohort 1 ^§^	Cohort 2 ^§^	*p*-Value
Cases, n	100	108	
Age (years), median (IQR)	65.0 (60.0–71.0)	65.0 (58.3–70.0)	0.408
Body Mass Index (kg/m^2^), median (IQR)	26.3 (24.3–28.7)	25.6 (23.3–28.2)	0.158
PSA (ng/mL), median (IQR)	8.5 (5.7–12.4)	8.7 (5.5–12.0)	0.706
Volume (mL), median (IQR)	54.0 (43.0–65.0)	47.0 (37.0–60.0)	0.010 **
Histopathological results of prostate biopsy			
Gleason score for biopsy, n/N			<0.010 **
6	21/100 (21.0%)	13/108 (12.0%)	
7a	34/100 (34.0%)	48/108 (44.4%)	
7b	18/100 (18.0%)	41/108 (38.0%)	
8	19/100 (19.0%)	3/108 (2.8%)	
9	8/100 (8.0%)	3/108 (2.8%)	
10	0/100 (0.0%)	0/108 (0.0%)	
Total number of biopsy cores, median (IQR)	12 (10–13)	12 (12–27)	<0.010 **
Number of positive biopsy cores, median (IQR)	4 (3–6)	4 (3–6)	0.973
Proportion of positive cores, median (IQR)	0.33 (0.21–0.50)	0.25 (0.13–0.42)	<0.010 **

IQR—interquartile range; PSA—prostate-specific antigen; (^§^) Cohort 1 was operated at the University Hospital in Munich and Cohort 2 was operated at the University Hospital in Freiburg. (**) *p* < 0.01.

**Table 2 cancers-14-05261-t002:** Outcomes of radical prostatectomy.

Parameter	Cohort 1	Cohort 2	
Cases, n	100	108	
Histopathological results of radical prostatectomy			
Gleason score for specimen, n/N			<0.010 **
6	11/100 (11.0%)	9/108 (8.3%)	
7a	44/100 (44.0%)	42/108 (38.9%)	
7b	20/100 (20.0%)	48/108 (44.4%)	
8	14/100 (14.0%)	5/108 (4.6%)	
9	11/100 (11.0%)	4/108 (3.7%)	
10	0/100 (0.0%)	0/108 (0.0%)	
Pathological T stage, n/N			<0.010 **
2a	2/100 (2.0%)	14/108 (13.0%)	
2b	3/100 (3.0%)	4/108 (3.7%)	
2c	52/100 (52.0%)	43/108 (4.0%)	
3a	20/100 (20.0%)	37/108 (3.4%)	
3b	23/100 (23.0%)	10/108 (9.2%)	
Pathological N stage, n/N			0.070
N0	86/100 (86.0%)	105/108 (97.2%)	
N1	9/100 (9.0%)	3/108 (2.8%)	
Nx	5/100 (5.0%)	0/108 (0.0%)	
Surgical margin status (T2 cohort), n/N			1.000
Negative	52/57 (91.2%)	56/61 (91.8%)	
Positive	5/57 (8.8%)	5/61 (8.2%)	
Surgical margin status (overall cohort), n/N			0.751
Negative	78/100 (78.0%)	89/108 (82.4%)	
Positive	18/100 (18.0%)	18/108 (16.7%)	
Data missing	2/100 (2.0%)	1/108 (0.9%)	
Duration of surgery (minutes), median, IQR	172 (150–196)	149 (134–174)	<0.010 **
Estimated blood loss ^$^ (mL), median, IQR	300 (200–400)	131 (99–188)	<0.010 **

IQR—interquartile range; (^$^) for Cohort 2 the blood loss was calculated by multiplying the perioperative difference of hemoglobin by the patient’s estimated blood volume (70 mL/kg bodyweight). (**) *p* < 0.01

**Table 3 cancers-14-05261-t003:** Hospital readmission rates and adverse events.

Parameter	Value—n/N
Intraoperative complications	0/108 (0.0%)
Conversion to open surgery	0/108 (0.0%)
Hospital readmission within 30 d	3/108 (2.8%)
Hospital readmission within 90 d	5/108 (4.6%)
Complications according to Clavien—Dindo	
1	13/108 (12.0%)
2	20/108 (18.5%)
3a	5/108 (3.7%)
Infectious Complications	
Wound infection	0/108 (0.0%)
Bacteremia	18/108 (16.7%)
Epididymitis	1/108 (0.9%)
Sepsis	1/108 (0.9%)
Urological and Surgical Complications	
Urinary retention	1/108 (0.9%)
Lymphocele with intervention	3/108 (2.8%)
Urinoma with intervention	1/108 (0.9%)
Ileus or revision with bowel involvement	0/108 (0.0%)
Blood transfusion	0/108 (0.0%)
Other Complications	
Acute pulmonary insufficiency	9/108 (8.3%)
Pneumonia	0/108 (0.0%)
Venous thrombosis	0/108 (0.0%)
Pulmonary embolism	0/108 (0.0%)
Myocardial infarction	0/108 (0.0%)
Acute kidney impairment	3/108 (2.8%)
Disorder of the liver	1/108 (0.9%)

## Data Availability

There is no data availability to report.

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
