# Peer review of "Development and Implementation of an Advanced Program for Robotic Treatment of Prostate Cancer—Is Surgical Quality Transferable?"

_cancers, 2022, doi:10.3390/cancers14215261_

Round 1

Reviewer 1 Report

Dear Authors:

The manuscript "Development and implementation of an advanced program for robotic treat-ment of prostate cancer – is surgical quality transferable?" by Sigle et al has demonstrated the quality of RARP is transferable if a surgeon with extensive specialist training and expertise is hired. I have just a few suggestions.

1. Some references or background information are missing. In introduction, please add more information about application of Da Vinci robots in several procedures to show the development and importance of research for Da Vinci surgery robot. (please cite: 1. Efficacy of da Vinci robot-assisted lymph node surgery than conventional axillary lymph node dissection in breast cancer - A comparative study. Int J Med Robot. 2021 Jul 16:e2307. doi: 10.1002/rcs.2307.  2.Robot-Assisted Minimally Invasive Breast Surgery: Recent Evidence with Comparative Clinical Outcomes. J Clin Med. 2022 Mar 25;11(7):1827. doi: 10.3390/jcm11071827.

2. If it it possible, please clarify the criteria of inclusion and exclusion of patients

Best,

Reviewer 2 Report

This is a e well described paper with interesting findings.

For the aim of the research it would have been more interesting analyzing a cohort of patients treated from a different surgeon trained by an expert surgeon. This might reveal in a better manner it is really possible to obtain some data on the trasferibility of surgical quality and about this is easier with robotic surgery compared to open or laparoscopic surgery.

Regarding oncological results the author might cite Gandi C, Totaro A, Bientinesi R, Marino F, Pierconti F, Martini M, Russo A, Racioppi M, Bassi P, Sacco E. A multi-surgeon learning curve analysis of overall and site-specific positive surgical margins after RARP and implications for training. J Robot Surg. 2022 Feb 28. doi: 10.1007/s11701-022-01378-w. Epub ahead of print. PMID: 35226289. regarding the maintenance of oncological quality from mentors to trainees.

Considering the title and the introduction, it might be interesting if the authors say something about the development of robotic surgical program in the new centre in order to find out if the good results do not rely only on the ability of the surgeon.

Reviewer 3 Report

This study is biased by few relevant limitations: the relatively small sample size, the fact that A comprehensive comparison between cohorts cannot be assessed due to the lack of data regarding the Hospital readmission rates and adverse events for Cohort 1; The lack of Trifecta achievement rate comparison and functional outcomes comparison between cohorts; the two cohorts were not matched by pT and pN stage and the different methods of blood loss estimation in the two cohorts. Nonetheless, the manuscript is well written, results are clear, and the topic is highly interesting. The overall quality of the manuscript can be further improved addressing the following comments:

1)      Was the prostate volume measured in the same way for both cohorts? Please specify. Weighting the surgical specimen is notably the most accurate way of estimating prostate volume.

2)      Was the Charlson comorbidity index comparable between the two cohorts?

3)      was the operative time measured as console time only ? please specify

4)      Anceschi et al. recently described a novel tool, namely “proficiency score” (PS) for evaluating the quality of RARP-LC and to analyze predictors of 1-year composite outcomes of RARP performed with two different surgical techniques by both trainers and their respective mentors at four tertiary-care centers. Results showed how trainers under supervision duplicate their respective mentors’ outcomes during Learning curve at high-volume centers irrespectively of surgical technique considered. Proficiency score achievement was the only independent predictors of 1-year trifecta outcomes in the trainer cohort. Please cite:

Anceschi U, Galfano A, Luciani L, et al. Analysis of predictors of early trifecta achievement after robot-assisted radical prostatectomy for trainers and expert surgeons: the learning curve never ends. Minerva Urol Nephrol. 2022 Apr;74(2):133-136.

Reviewer 4 Report

The authors should be complimented for their scientific and surgical skills. The article demonstrates that a center starting a new procedure would better hire somebody expert. No surprise, this is a somehow intuitive concept. However, the article has a sound design and it is well written. Results are excellent: the authors might consider to add the outcomes of the surgeon in his former center, in order to quantify precisely the transferability of a standardized procedure from a high volume center to a starting center.

Reviewer 5 Report

Dear Editor and Author, In my opinion, whole paper does not raise any substantive doubts.
It is written in clear and transparent scientific language.
All abbreviations are clearly explained.
The methodology, results and conclusions are detailed. As a robotic surgeon and robotic proctor I fully agree that the full
pre-training increases the treatment result, deceases complications and
shortens the operation time even in the case of more locally advanced cancer.
I fully recommend accepting the paper for publishing in Cancers. Regards

Round 2

Reviewer 1 Report

suggest to publish